# An Epidemiological Study to Investigate Links between Atmospheric Pollution from Farming and SARS-CoV-2 Mortality

**DOI:** 10.3390/ijerph19084637

**Published:** 2022-04-12

**Authors:** Paolo Contiero, Alessandro Borgini, Martina Bertoldi, Anna Abita, Giuseppe Cuffari, Paola Tomao, Maria Concetta D’Ovidio, Stefano Reale, Silvia Scibetta, Giovanna Tagliabue, Roberto Boffi, Vittorio Krogh, Fabio Tramuto, Carmelo Massimo Maida, Walter Mazzucco

**Affiliations:** 1Environmental Epidemiology Unit, Fondazione IRCCS Istituto Nazionale dei Tumori, 20133 Milan, Italy; paolo.contiero@istitutotumori.mi.it (P.C.); martina.bertoldi@istitutotumori.mi.it (M.B.); 2International Society of Doctors for Environment (ISDE), 52100 Arezzo, Italy; 3UOC Qualità dell’Aria, ARPA Sicilia, 90146 Palermo, Italy; abita@arpa.sicilia.it; 4Reporting Ambientale, Salute e Ambiente, ARPA Sicilia, 90146 Palermo, Italy; gcuffari@arpa.sicilia.it; 5Inail-Dipartimento di Medicina, Epidemiologia, Igiene del Lavoro ed Ambientale, Monte Porzio Catone, 00078 Rome, Italy; p.tomao@inail.it (P.T.); m.dovidio@inail.it (M.C.D.); 6Laboratorio Tecnologie Diagnostiche Innovative Area Biologia Molecolare, Istituto Zooprofilattico Sperimentale della Sicilia, Via Rocco Dicillo 3, 90129 Palermo, Italy; stefano.reale@izssicilia.it (S.R.); silvia.scibetta@hotmail.it (S.S.); 7Cancer Registry Unit, Fondazione IRCCS Istituto Nazionale dei Tumori, 20133 Milan, Italy; giovanna.tagliabue@istitutotumori.mi.it; 8Respiratory Disease Unit, Fondazione IRCCS Istituto Nazionale Tumori, 20133 Milan, Italy; roberto.boffi@istitutotumori.mi.it; 9Epidemiology and Prevention Unit, Fondazione IRCCS Istituto Nazionale dei Tumori, 20133 Milan, Italy; vittorio.krogh@istitutotumori.mi.it; 10Department of Health Promotion, Mother and Child Care, Internal Medicine and Medical Specialties (PROMISE) “G. D’Alessandro”—University of Palermo, 90127 Palermo, Italy; fabio.tramuto@unipa.it (F.T.); carmelo.maida@unipa.it (C.M.M.); walter.mazzucco@unipa.it (W.M.); 11Regional Reference Laboratory of West Sicily for the Emergency of COVID-19, Clinical Epidemiology Unit, University Hospital “Paolo Giaccone”, 90127 Palermo, Italy; 12Division of Biostatistics and Epidemiology, Cincinnati Children’s Hospital Medical Centre, Cincinnati, OH 45229, USA

**Keywords:** particulate matter, SARS-CoV-2, COVID-19, mortality, pollution, ammonia, farming, livestock, import and export, gross domestic product

## Abstract

Exposure to atmospheric particulate matter and nitrogen dioxide has been linked to SARS-CoV-2 infection and death. We hypothesized that long-term exposure to farming-related air pollutants might predispose to an increased risk of COVID-19-related death. To test this hypothesis, we performed an ecological study of five Italian Regions (Piedmont, Lombardy, Veneto, Emilia-Romagna and Sicily), linking all-cause mortality by province (administrative entities within regions) to data on atmospheric concentrations of particulate matter (PM_2.5_ and PM_10_) and ammonia (NH_3_), which are mainly produced by agricultural activities. The study outcome was change in all-cause mortality during March–April 2020 compared with March–April 2015–2019 (period). We estimated all-cause mortality rate ratios (MRRs) by multivariate negative binomial regression models adjusting for air temperature, humidity, international import-export, gross domestic product and population density. We documented a 6.9% excess in MRR (proxy for COVID-19 mortality) for each tonne/km^2^ increase in NH_3_ emissions, explained by the interaction of the period variable with NH_3_ exposure, considering all pollutants together. Despite the limitations of the ecological design of the study, following the precautionary principle, we recommend the implementation of public health measures to limit environmental NH_3_ exposure, particularly while the COVID-19 pandemic continues. Future studies are needed to investigate any causal link between COVID-19 and farming-related pollution.

## 1. Introduction

According to the World Health Organization, on 24 January 2022, worldwide there were 349,641,119 confirmed cases of SARS-CoV-2 infection (COVID-19) and 5,592,266 deaths (1.60% of confirmed cases) [1]. The corresponding figures for Italy were 9,923,678 confirmed cases and 143,523 (1.45%) deaths [1]. These are likely to be underestimates [2].

Several studies have investigated effects of environmental and meteorological factors on the dissemination and severity of viral respiratory infections [3,4,5,6]. A 2018 paper provided evidence that air pollution from coal burning had exacerbated the “Spanish flu” pandemic of 1918 [7]. Although most COVID-19 patients develop mild or no symptoms, more severe and life-threatening symptoms such as pneumonia (often leading to acute respiratory distress syndrome), vascular inflammation and thrombosis, myocarditis, and cardiac arrhythmia, also occur, typically associated with excessive inflammation and cytokine storm [8].

Long-term and short-term exposure to atmospheric particulate matter ≤10 μm (PM_10_) and ≤2.5 μm (PM_2.5_) have been linked to respiratory and cardiovascular events [9]. A recent US study found that an increase of just 1 μg/m^3^ in long-term exposure to PM_2.5_ was associated with an 8% increase in the COVID-19 death rate [10]. Studies in Italy have identified associations between COVID-19 and COVID-19-related death and exposure to PM_10_ and PM_2.5_ [11,12,13,14], while tropospheric nitrogen dioxide (NO_2_) in northern Italy was associated with levels of SARS-CoV-2 infection [15]. The COVID-19 pandemic in Italy started in the Po Valley of northern Italy—one of the most polluted areas in the world [16]—where intensive livestock rearing and heavy use of fertilizers make major contributions to atmospheric pollution [17]. In December 2019 in the Po Valley Regions of Lombardy, Piedmont, Emilia-Romagna and Veneto, there were, respectively, 1,543,639; 824,801; 627,627 and 824,112 cattle, and 3,984,633; 1,121,723; 1,377,527 and 717,557 pigs. In comparison, the island Region of Sicily had 387,619 cattle and 45,152 pigs [18]. As regards nitrogen fertilizer use, the 2019 figures (tonne/km^2^/year) were 12.92 in Lombardy, 8.27 in Piedmont, 18.64 in Veneto and 19.50 in Emilia-Romagna, compared to 5.20 in Sicily [19].

Moreover, many international research groups have conducted descriptive analyses evaluating the impact of the COVID-19 pandemic on air quality during lockdown, mostly in urban areas. They reported a temporary improvement in air quality and a decrease in atmospheric emissions from the transportation sector [20,21].

Agricultural activity produces many pollutants, mainly NH_3_ and PM_10_ and PM_2.5_, also, whereas NH_3_ can be used as a proxy for all the pollutants produced by intensive agricultural activities [22,23,24]. In Italy, agriculture is the main source of NH_3_ emissions, with an estimated 362.18 kilotonnes/year, accounting for 94.3% of the total [22]. In Lombardy in 2017 (latest year for which figures are available) the proportion of NH_3_ emissions from agriculture varied by province (68–99%) [25].

Analysis of all-cause mortality data indicates that COVID-19-related deaths are underestimated [2,26]. A study that analysed 22 countries reported that in Italy COVID-19-related deaths were underestimated by 30% with similar underestimates in several other countries [27]. For this reason, all-cause mortality may represent the epidemiologic indicator of choice for assessing the mortality impact of COVID-19 [2,28,29] from which excess deaths can be estimated as proxy of COVID-19 mortality [28]. Moreover, factors such as mobility, economic activities, social interactions, level of globalization of an investigated area, and total international import and export (used as synthetic parameter), have been considered important drivers of virus spread [30].

We hypothesized that long-term exposure to farming-related air pollutants might predispose to an increased risk of COVID-19-related death. The aim of this study was, therefore, to investigate if exposure to farming-related atmospheric pollution may worsen the effect of SARS-CoV-2 on mortality. To this end, we performed an ecological study to find out whether the results justified additional analytic studies to identify modifiable risk factors to treat for limiting the effect of the COVID-19 on mortality. We included in our model the effect of total international import-export and of gross domestic product as a proxy of economic development by province.

## 2. Materials and Methods

### 2.1. Study Design

We conducted an ecological study to investigate whether interaction between SARS-CoV-2 infection and exposure to farming-related air pollutants worsened the effect of SARS-CoV-2 on mortality at the level of the provinces (administrative entities within regions) in the Italian Regions of Lombardy, Emilia-Romagna, Piedmont, Veneto and Sicily (see Figure 1). We investigated the period of the first COVID-19 wave (March–April 2020) in comparison with the same two-month period in the years 2015 to 2019. Mortality differences between these two periods were considered a proxy for mortality due to COVID-19.

### 2.2. Pollutant Exposure

We assessed atmospheric concentrations of PM_2.5_, PM_10_, NH_3_ and NO_2_. The first three pollutants were considered because linked to farming, while NO_2_ was included because it could be a confounder or effect modifier. Data on atmospheric PM_2.5_, PM_10_ and NO_2_ levels were accessed from the national monitoring system coordinated by the Italian National Institute for Environmental Protection and Research (ISPRA) [31]. We used data from the monitoring stations managed by Regional Environmental Protection Agencies (ARPAs) provided at a province level. To estimate long-term exposure, we averaged daily concentrations from 2016 to 2019. We excluded from the analysis two provinces for which PM_10_ and NO_2_ annual concentrations were not available for the period under study. We did not use 2020 data, as they could have biased estimates because of changes in pollutant levels caused by lockdown [20,21,32,33]. As NH_3_ concentrations provided by from ARPAs are monitored by too few stations to be representative of the areas being studied, we estimated concentrations of atmospheric NH_3_ at the province level as well [25,34,35,36,37,38]. More in depth, ARPAs estimate NH_3_ emissions for each province and for each known source (e.g., pigs) by multiplying the average number pigs per year present in the province by an estimate of the average production by each animal (as kg NH_3_/animal/year). A similar method is applied to each type of livestock present and also other NH_3_ sources (e.g., fertilizer use). Total NH_3_ emissions were obtained by summing estimated emissions from all known sources. We next derived provincial exposure indexes from each ARPA estimate by dividing the total estimated quantities of emissions (tonne/year) by the area (km^2^) of the province, thereby taking into account variations in the province area. We used data for the latest available year as proxy for long-term exposure: 2017 for Lombardy and Emilia-Romagna (except for one province for which data were for 2016), 2015 for Piedmont and Veneto and 2012 for Sicily. Between the two extremes of the latest available years, 2012 and 2017, a 4.77% decrease in NH_3_ was observed in the agricultural sector throughout Italy [22].

Backes et al. [39] showed that the atmospheric concentration patterns of NH_3_ are in line with NH_3_ emission patterns, indicating the provincial exposure indexes are acceptable indicators of atmospheric NH_3_. To further investigate exposure indexes as indicators of atmospheric pollution levels for NH_3_, we downloaded available data from the Italian National Statistics Institute (ISTAT) [18,19] on numbers of cattle and pigs and fertilizer used per km^2^ by region. For each one of the twenty Italian regions, we downloaded data about NH_3_ emissions from ISPRA [22], and we built regional exposure indexes in the same way we did for provinces. We then assessed correlations (Pearson’s r) between NH_3_ emissions and cattle numbers, pig numbers and fertilizer use per km^2^ by region. We could only compare data at the regional level because of the unavailability of data on numbers of pigs and cattle and nitrogen fertilizer use at the provincial level.

### 2.3. Meteorological Variables

We hypothesized that temperature influenced COVID-19 rates over the short-term and so might have an effect on mortality. We therefore used temperature and humidity, as measured by ARPA monitoring stations in the main town of each province, as proxies for the average temperature and average relative humidity in each province, in March and April in 2020, and in March–April 2019 also.

### 2.4. Additional Covariates

Social deprivation is also linked to SARS-CoV-2 infection and death [40]. However, social deprivation indexes at the provincial level were not available, so we assumed that the gross domestic product per capita was a rough proxy for social deprivation. International trade has been hypothesized to be linked to COVID-19 spread [30], and therefore, we used these variables provided by ISTAT [18,19] to create an indicator as the ratio between the sum of the international import and export activities in Euros by provinces, divided by the size of the provincial population. We also introduced in the model the gross domestic product per capita (GDPc) by province (ISTAT), as a proxy indicator of population wealth and of socio-economic status.

High body mass index (BMI) and diabetes are recognized as major risk factors for SARS-CoV2 death [41,42] and could bias estimates of links between farming-related pollutants and SARS-CoV-2 mortality. Provincial level data were unavailable, so we accessed regional level data from ISTAT [43] and assessed correlations (Pearson’s r) with NH_3_ to provide indications as to the possible effects of BMI and diabetes on our findings. 

Variation in lockdown periods across provinces could also bias the results. However, in Italy during the first COVID-19 wave, days of lockdown were closely similar throughout the country. Similarly, regulations to limit contagion (mask-wearing, social distancing) applied to the whole country during the two-month study period.

### 2.5. Outcome

All-cause daily mortality data provided by ISTAT [44] were the primary study outcome. As mortality data are available by municipality (administrative entities within provinces), we calculated total mortality by province, for March-April 2020, as the sum of the daily death counts in all the municipalities in each province. We also estimated average mortality in March-April of 2015–2019, as the average over the five years of the sum of daily death counts in all the municipalities of each province.

### 2.6. Statistical Methods 

We first sought correlations (Pearson’s r) between levels of the atmospheric pollutants being studied (PM_2.5_, PM_10_, NH_3_ and NO_2_). We next analyzed associations between changes in total mortality from March–April 2015–2019 to 2020 and atmospheric pollutant levels, humidity, temperature, and population density (study covariates). To this end, we used negative binomial regression models, that included a population size offset, and which estimated mortality rate ratios (MRRs) in relation to study covariates. MRR is the ratio of the mortality rate for a specific value of a covariate relative to reference. Thus, for the covariate “period” MRR was the mortality in March-April 2020 relative to mortality in March–April in 2015–2019 and is a good proxy for the increase in mortality due to COVID-19. We were particularly interested in MRRs for the interaction between period and the other covariates (mainly pollutant levels) as these are an estimate of the mortality associated with the pollutant over and above that due to COVID-19 alone.

We first ran basic models that included period, one covariate and an interaction term between period and covariate. We next ran complete models that included all covariates and all interaction terms between period and covariates.

Information on PM_10_ levels was available for all provinces, while PM_2.5_ data were unavailable for four provinces. We therefore ran two sets of models, one that included all 43 provinces but excluded PM_2.5_ and another that included PM_2.5_ and the 39 provinces for which PM_2.5_ was available. The R statistical package [45], version 4.0.2, was used to perform the analyses.

## 3. Results

As compared to mortality documented in March–April 2015–2019, mortality in March–April 2020 varied from a 1.8% decrease in Agrigento Province (Sicily) to a 364.3% increase in Bergamo (Lombardy) (data not shown). 

Mean values of atmospheric PM_10_, PM_2.5_ and NO_2_ across provinces were 28.03, 17.58 and 29.84 μg/m^3^, respectively; the PM_10_ ranged from 15.25 (Enna, Sicily) to 37.25 μg/m^3^ (Padua, Veneto), the PM_2.5_ ranged from 8.0 (Enna, Sicily) to 28.75 μg/m^3^ (Padua, Veneto), and the NO_2_ ranged from 4.0 (Agrigento, Sicily) to 49.0 μg/m^3^ (Monza Brianza, Lombardy) (data not shown).

As regards emissions, mean across-province exposure index estimates for NH_3_ was 2.41 tonne/km^2^/year. The NH_3_ exposure index ranged from 0.12 (Verbano-Cusio-Ossola, Piedmont) to 10.3 tonne/km^2^/year (Cremona, Lombardy) (data not shown).

Table 1 reports Pearson correlation coefficients for PM_10_, PM_2.5_, NO_2_, NH_3,_ import-export and gross domestic product.

Table 2 presents modelling results for all 43 provinces included in the study: the MRRs represent percentage increases in overall death rate in relation to increments of 1 μg/m^3^ in PM_10_, and NO_2_, 1 tonne/km^2^ increments in NH_3_, a 1 °C increment in atmospheric temperature, a 1% increment in relative humidity, a one unit increase in population density, a 1000 Euros increment in international import-export per capita and a 1000 Euros increment in GDPc (Gross Domestic Product per capita).

The basic regression models identified a significantly increased in MMRs for interactions between period and PM_10_ (MRR: 1.030; 95%CI: 1.009–1.052), NO_2_ (MRR: 1.017; 95%CI: 1.001–1.032) and NH_3_ (MRR: 1.093; 95%CI: 1.039–1.149), indicating that high levels of these pollutants were significantly linked to an increase in total mortality over and above that due to period (proxy for presence of COVID-19). None of the individual covariates was significantly associated with MRR.

In the complete model, MRRs for the interaction between NH_3_ and period (MRR: 1.069; 95%CI: 1.006–1.136) remained significant, indicating that an increase of 1 tonne/km^2^/year in NH_3_ emissions was significantly associated with a 6.9% increase in all-cause death over and above that associated with period (proxy for COVID-19). MRRs for interaction of period with PM_10_ and NO_2_ were reduced, compared to the basic model, and became non-significant.

Table 3 shows results for the 39 provinces for which PM_2.5_ data were available. The MRRs were similar to those obtained for all 43 provinces. As regards the complete model, only the MRR for interaction between period and NH_3_ was significant (1.072; 95%CI: 1.001–1.151) corresponding to a 7.2% increase (over and above that for period alone) in mortality for each tonne/km^2^/year increase in NH_3_ emissions.

Correlation analysis found the following correlations: 0.92 between provincial exposure index for NH_3_ and number of pigs per km^2^; 0.98 between provincial exposure index for NH_3_ and number of cattle per km^2^; 0.72 between the provincial exposure index for NH_3_ and fertilizer use per km^2^. These findings suggest that emission levels are a reasonable surrogate for atmospheric levels.

In addition, correlations were: −0.87 between BMI and the regional exposure index for NH_3_ and −0.77 between diabetes and the regional exposure index for NH_3_. Given the strength of the association of COVID-related mortality increase (6.9%) due to NH_3,_ it is unlikely that BMI and diabetes would have had biased this association.

## 4. Discussion

Using basic models, which excluded most covariates, we found that higher NH_3_ emissions—whose major source is agriculture—were associated with all-cause mortality increases in 2020 compared to previous years, over and above those due to period (proxy for COVID-19–related mortality). Higher PM_10_, PM_2.5_ and NO_2_ levels were also associated with all-cause mortality increases over and above those due to period.

Using complete models, which included all other studied pollutants as covariates, higher NH_3_ levels remained associated with a significant all-cause mortality increase: a 1-tonne/km^2^ increase in NH_3_ was associated with a 6.9% increase in mortality over and above that due to period. The fact that interactions of period with NO_2_, PM_10_ and PM_2.5_ were not associated with excess mortality in the complete models is likely due to the fact that the levels of these pollutants correlated highly with each other.

A major study assumption was that the provincial exposure index for NH_3_ based on pollutant emissions is a good proxy for the levels of that pollutant in the atmosphere. This assumption is supported by the study of Backes et al. [39], which showed that atmospheric concentration patterns of NH_3_ are in line with NH_3_ emission patterns. To further investigate provincial exposure indexes as indicators of atmospheric pollution levels, we assessed correlations (Pearson’s r) between NH_3_ emissions and cattle numbers, pig numbers and nitrogen fertilizer use per km^2^ by region (provincial data unavailable): we found high or good correlations (0.98, 0.92 and 0.72, respectively), indicating that estimated emissions are consistent with the emission sources present and suggesting that these estimated emissions (standardized by area size) are a reasonable proxy for atmospheric NH_3_ concentrations.

It is known that BMI and diabetes are major risk factors for COVID-19 infection and death. However, provincial data for BMI and diabetes were not available and could not be included in our models as covariates. We indirectly assessed whether the association between farming-related pollutants and SARS-CoV-2 mortality were biased by BMI and diabetes by estimating correlations between BMI and NH_3_ and between diabetes and NH_3_ at the regional level. As noted, given the strength of the association of the COVID-related mortality increase (6.9%) due to NH_3_, it is unlikely that BMI, diabetes or income would have exerted a major bias on this association. Any influence of population density and educational level on associations was taken into account by entering these as covariates in the models.

To our knowledge, this is the first study to investigate relationships between farming-related atmospheric pollution and COVID-19-related mortality. However, a 2020 study conducted in Italy found that the number of SARS-CoV-2 infections per unit of population depended on the type of rural landscape [46]. The study classified rural landscapes into (A) urban and periurban with high-intensity agriculture; (B) high-intensity agriculture; (C) medium-intensity agriculture (hilly areas); (D) low-intensity agriculture (high hills and mountains). They found that areas of less intensive agriculture (C and D) had fewer COVID-19 cases (per 10,000 inhabitants) than areas of intensive agriculture (A and B). Notwithstanding the markedly different methodologies, both our study and the mentioned one [46] identified intensive farming as a risk factor for COVID-19. Another study conducted in Italy [47] linked the short-term exposure to some atmospheric pollutants and hypothesized also a possible role for NH_3_ associated with COVID-19 spread. Despite the differences in outcome—spread versus mortality and short-term versus long-term one—our study and the latter one seem to point in the same direction.

A study conducted in Italy by Bontempi et al. [30] identified international import-export as the major driver for COVID-19 spread. In our study, international import-export in the complete analysis showed a possible (although not statistically significant) association, but less than NH_3_ and comparable to NO_2_. One explanation could be that international import-export may influence the place where SARS-CoV-2 starts in a country (in Italy, for example, the Codogno municipality), but once SARS-CoV-2 has started, then it rapidly spreads everywhere, independent of international import-export. A difference between our study and the one by Bontempi et al. was the different metrics used to take import-export into account. When developing our study, we used a variable created by dividing the total import-export by provincial population size instead of the rough total amount of import-export.

Similarly, the 2020 Lancet Countdown report highlighted a link between SARS-CoV-2 infection, environmental degradation and climate change and identified agriculture as one of the main sources of pollutant emissions [48].

Other studies have reported associations between exposure to livestock farming and respiratory diseases. A study in the Netherlands found that people living in high-livestock-density areas had a significantly higher prevalence of pneumonia than those living in low-density areas [49]. A German study [50] found that people exposed to higher levels of NH_3_ had poorer pulmonary health and were more likely to be sensitized to ubiquitous allergens than persons with lower NH_3_ exposure. A study performed in an agricultural region of Washington State (US) found that industrial-scale animal feeding operations were associated with higher daily outdoor NH_3_ levels and that forced expiratory volume in one second was lower with each interquartile increase in the previous day’s NH_3_ exposure; however, no association with asthma symptoms was observed [51]. The European Academy of Allergy and Clinical Immunology position paper [52] highlighted that increased risks of developing respiratory diseases and allergies were associated with animal farming and emphasized the need to protect workers from such risks.

The data provided by the above-cited studies [49,50,51,52] suggest that a link between excess COVID-19-related mortality and farming-related atmospheric pollutants is biologically plausible. The specific association we found between COVID-19-related mortality and NH_3_ might be explained by the hypothesis that atmospheric NH_3_ leads to the formation of an alkaline aerosol which triggers a conformation change in the SARS-CoV-2 spike that facilitates fusion of the viral envelope with the plasma membrane of target cells [53]. It is noteworthy that clusters of SARS-CoV-2 infection have been reported in association with slaughterhouses [54], which are known to have elevated levels of NH_3_ [55].

We used change in all-cause total mortality in 2020 as a proxy for SARS-CoV-2 mortality, and to our knowledge, our study is the first to do this. A strength of this approach is that it avoids loss of deaths due to underreporting of the SARS-CoV-2 mortality [26]. However, the pandemic may have stretched the health resources, so that more people than usual may have died from non-COVID-19 causes, somewhat inflating the all-cause mortality.

The main limitations of the study are that, for the atmospheric pollutants that were measured, we used levels measured at the monitoring station in the chief town of the province as a proxy for exposure of the entire provincial population. In addition, since no measured levels were available for NH_3_, we used estimated emissions at the provincial level obtained from the regional environmental protection agencies as proxies for exposure of the provincial populations.

## 5. Conclusions

Although this study was ecological in design and could not identify causal links between farming-related atmospheric pollutants and COVID-19 mortality, if we follow the precautionary principle, our findings support the implementation of public health measures to limit environmental NH_3_ exposure from the agricultural sector, particularly while the COVID-19 pandemic continues. Of interest, covering tanks containing slurry can speedily reduce NH_3_ emissions from agriculture [56,57]. However, medium-term structural changes are required to reduce farming-related pollution. Consumption of meat, particularly red meat, is recognized as a threat to human health and well-being in part because of the effect of livestock rearing on the environment [48]. As the desire to consume meat encourages the use of intensive farming, resulting in increased emission of farming-related pollutants, we need to move towards a more sustainable agriculture with reduced livestock raising [46,48]. Lastly, we note that intensive agriculture is a major cause of biodiversity loss worldwide, which has been implicated in the emergence of SARS-CoV-2 and other viruses [48,56].

Future studies are needed to investigate any causal link between COVID-19 and farming-related pollution.

## Figures and Tables

**Figure 1 ijerph-19-04637-f001:**
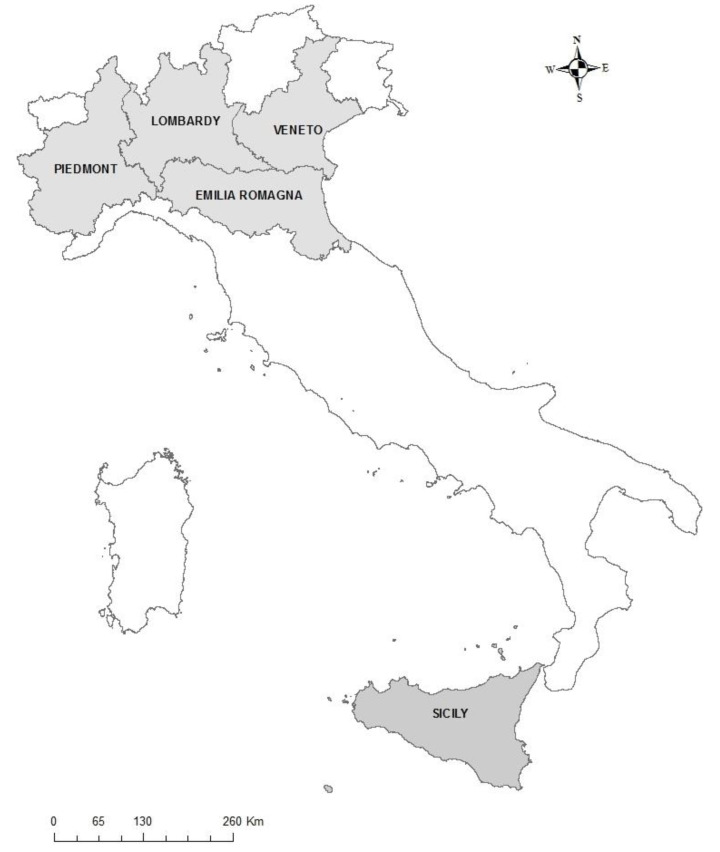
Location of the Italian Regions of Lombardy, Emilia-Romagna, Piedmont, Veneto and Sicily.

**Table 1 ijerph-19-04637-t001:** Pearson’s correlation coefficients between study variables.

Air Pollutant	PM_10_	PM_2.5_	NO_2_	NH_3_	Import-Export	GDPc
PM_10_	1.00	0.89 (0.83–0.93)	0.65 (0.50–0.76)	0.47 (0.27–0.62)	0.49 (0.30–0.64)	0.45 (0.27–0.61)
PM_2.5_	0.89 (0.83–0.93)	1.00	0.60 (0.53–0.78)	0.47 (0.28–0.63)	0.52 (0.39–0.67)	0.52 (0.31–0.65)
NO_2_	0.65 (0.50–0.76)	0.60 (0.53–0.78)	1.00	0.14 (–0.08–0.35)	0.20 (0.07–0.48)	0.41 (0.22–0.58)
NH_3_	0.47 (0.27–0.62)	0.47 (0.28–0.63)	0.14 (–0.08–0.35)	1.00	0.53 (0.35–0.68)	0.35 (0.15–0.52)
Import-export	0.49 (0.30–0.64)	0.52 (0.39–0.67)	0.20 (0.07–0.48)	0.53 (0.35–0.68)	1.00	0.59 (0.44–0.72)
GDPc	0.45 (0.27–0.61)	0.52 (0.31–0.65)	0.41 (0.22–0.58)	0.35 (0.15–0.52)	0.59 (0.44–0.72)	1.00

**Table 2 ijerph-19-04637-t002:** Mortality rate ratios (MRRs) * with 95% confidence intervals (CI) for all 43 Italian provinces included in the study.

Variable	MRR * (95%CI)	
	Basic Model ^§^	Complete Model ^‡^
PM_10_	0.997 (0.982–1.012)	1.010 (0.990–1.031)
PM_10_ and period interaction	1.030 (1.009–1.052)	0.991 (0.962–1.020)
NO_2_	0.993 (0.983–1.003)	0.992 (0.980–1.005)
NO_2_ and period interaction	1.017 (1.001–1.032)	1.014 (0.996–1.032)
NH_3_	0.991 (0.958–1.028)	0.984 (0.943–1.029)
NH_3_ and period interaction	1.093 (1.039–1.149)	1.069 (1.006–1.136)
Temperature	0.981 (0.898–1.073)	1.000 (0.921–1.085)
Temperature and period interaction	0.937 (0.815–1.077)	0.921 (0.816–1.039)
Humidity	1.004 (0.990 –1.020)	1.003 (0.987–1.021)
Humidity and period interaction	0.997 (0.979–1.016)	0.998 (0.978–1.020)
Population density	1.000 (0.999–1.001)	1.000 (0.999–1.001)
Population density and period interaction	1.000 (0.999–1.001)	1.000 (0.999–1.001)
Import-export	0.998 (0.990–1.006)	0.999 (0.988–1.010)
Import-export and period interaction	1.022 (1.010–1.034)	1.014 (0.998–1.029)
GDP-pc	0.995 (0.983–1.007)	1.003 (0.988–1.017)
GPD-pc and period interaction	1.023 (1.005–1.042)	0.998 (0.972–1.025)
Period (2020 vs. 2015–2019)	1.771 (1.551–2.021)	3.017 (0.211–42.661)

* MMRs are percentage increases in overall death rate associated with 1 μg/m^3^ increase in atmospheric PM10, 1 μg/m^3^ increase in atmospheric NO_2_, 1 tonne/year/km^2^ increases in emissions of NH_3_, 1 °C increase in atmospheric temperature, 1% increase in relative humidity, one unit increase in population density, a 1000 Euros increment in international import–export per capita and a 1000 Euros increment in Gross Domestic Product per capita. ^§^ Basic models include only period, a single covariate and its interaction with period. ^‡^ Complete models include all covariates together with their interactions with period.

**Table 3 ijerph-19-04637-t003:** Mortality rate ratios (MRRs) * with 95% confidence intervals (CI). Analyses restricted to the 39 Italian provinces for whichPM_2.5_ data were available.

Variable	MRR * (95%CI)	
	Basic Model ^§^	Complete Model ^
PM_10_	0.997 (0.982–1.012)	1.009 (0.977–1.043)
PM_10_ and period interaction	1.026 (1.004–1.049)	1.002 (0.957–1.050)
PM_2.5_	0.993 (0.975–1.012)	1.004 (0.960–1.050)
PM_2.5_ and period interaction	1.033 (1.005–1.061)	0.974 (0.914–1.038)
NO_2_	0.992 (0.982–1.003)	0.990 (0.976–1.004)
NO_2_ and period interaction	1.016 (1.001–1.033)	1.020 (1.000–1.041)
NH_3_	0.991 (0.956–1.029)	0.973 (0.924–1.025)
NH_3_ and period interaction	1.082 (1.027–1.139)	1.072 (1.001–1.151)
Temperature	0.980 (0.883–1.087)	1.026 (0.919–1.144)
Temperature and period interaction	1.010 (0.858–1.187)	0.914 (0.776–1.076)
Humidity	1.006 (0.989–1.022)	1.005 (0.987–1.024)
Humidity and period interaction	1.003 (0.983–1.023)	1.000 (0.978–1.023)
Population density	1.000 (0.999–1.001)	1.000 (0.999–1.001)
Population density and period interaction	1.000 (0.999–1.001)	1.000 (0.999–1.001)
Import-export	0.998 (0.989–1.007)	1.002 (0.989–1.015)
Import-export and period interaction	1.022 (1.009–1.035)	1.016 (0.998–1.034)
GDP-pc	0.993 (0.979–1.007)	1.001 (0.981–1.015)
GPD-pc and period interaction	1.015 (0.994–1.036)	0.994 (0.966–1.022)
Period (2020 vs. 2015-2019)	1.846 (1.611–2.116)	3.287 (0.144–73.775)

* MMRs: percentage increases in overall death rate associated with 1 μg/m^3^ increase in atmospheric particulate matter (PM_2.5_ or PM_10_), 1 μg/m^3^ increase in atmospheric NO_2_, 1 tonne/km^2^/year increases NH_3_, 1 °C increase in atmospheric temperature, 1% increase in relative humidity, one unit increase in population density, a 1000 Euros increment in international import–export per capita and a 1000 Euros increment in Gross Domestic Product per capita. ^§^ Basic models include only period, a single covariate and its interaction with period. ^ Complete models include all covariates and their interactions with period.

## Data Availability

This study used only aggregated data available on web-sites accessible to the public.

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
