# Peer review of "An Epidemiological Study to Investigate Links between Atmospheric Pollution from Farming and SARS-CoV-2 Mortality"

_ijerph, 2022, doi:10.3390/ijerph19084637_

Round 1

Reviewer 1 Report

Dear Authors,

The article "An Epidemiological Study to Investigate Links Between Atmospheric Pollution From Farming and SARS-CoV-2 Mortality", is very interesting and well explained.

Just a few recommendations:
In the "Study Design", maybe add a map showing the study sites.
In the "Meteorological Variables" why wasn't precipitation added? Air pollutants are influenced by precipitation.
In the results show the averages with their standard deviations.

The authors used "binomial regression models", only as a recommendation for future studies try to use "geographical weighted regression" and compare the results. 

Reviewer 3 Report

Prior to raising comments, I would like to thank authors for conducting this study. The study has been well designed and conducted as well as scholarly way presented the findings. The following is some comments to be addressed before publication.

Please discuss the effect of COVID-19 on ambient air quality all over the world in the introduction section based on the following review papers.

https://www.mdpi.com/1660-4601/19/4/1950

https://online.ucpress.edu/elementa/article/9/1/00176/116616/The-global-impacts-of-COVID-19-lockdowns-on-urban

Author Response

This manuscript is a resubmission of an earlier submission. The following is a list of the peer review reports and author responses from that submission.

Round 1

Reviewer 1 Report

This is a very meaningful study: to study the correlation between pollutants produced by agricultural activities and the rate of death. Because there is no direct relationship between agricultural pollution and neo-coronavirus mortality, the authors analyzed the comparison of inter-annual mortality rates in 2015-2019 to illustrate the contribution of pollutants from agricultural production to mortality. The research methods and results are basically reliable and have some guiding significance for the prevention and treatment of the new coronavirus.

However, there are still some issues that need to be clarified:

  1. Abstract line 40: "all produced by agricultural activities" is not appropriate, it is recommended to change to "mainly produced by agricultural activities".
  2. Line 112: Missing a space, should be "NH3 concentration..."
  3. Line 115-119, it is only an example to illustrate the ARPA’s activity to estimate emission of NH3, however the authors did not clearly explain the method of NH3 calculation and the corresponding coefficients, please explain the exact calculation steps.
  4. Line 161, there is no Table S1 in the manuscript.
  5. Lines 176-181: It should be the correlation between air pollution components and the number of deaths, rather than between different pollution components. It is recommended that the author change this paragraph to the correlation between the air pollutions and the number of deaths.
  6. What do the variables in Table 1 mean, such as "PM10" and "PM10 and period interaction", is it the MRR (95%CI) in 2019 and in 2020, respectively?
  7. Reference: the journal name abbreviation needs to be unified.

Author Response

Dear Reviewer,

we thank you for your suggestions.
We have proceeded to answer all your questions in red.

We hope the revised paper may now be suitable for publication in the International Journal of Environmental Research and Public Health.

Your sincerely,

Dr. Alessandro Borgini

Reviewer 2 Report

A better understanding of the environmental influences on the COVID pandemic are critically important for designing interventions to decrease societal disease burden and to better prepare for future viral pandemics. The unique finding of NH3 associated with excess deaths during the COVID epidemic is an important one to get out into the literature such that these findings can be confirmed or refuted with more rigorous study designs. Use of excess mortality is an appropriate outcome, as justified by the authors.

General Comments:

-Please justify the use of source emissions data as a surrogate for NH3 air pollution. Do number of pigs correlate with provincial level NH3 air pollution levels in the referenced studies. In other words, has your methodology been validated in other studies? Reference, please.

The correlation analysis might be better presented as a figure, correlation matrix

-Please present a descriptive table of provinces (with more detail than table S1) also including Emissions by province, Air pollution concentrations by province, and  with 2015-19  mortality compared to 2020 mortality, and the % excess mortality, by province.

-The main limitation of this study, as thoroughly disclosed by the authors, is its ecologic design. As it stands, after reading this article, I still have no idea if NH3 truly increases the risk for COVID mortality.  Are there additional provincial-level covariates that can be included in the models to make these analyses more robust? Could the authors include provincial level census data on mean household income and household education to account for socioeconomic status that could be driving COVID related mortality, as we know that health care disparities are associated with increased COVID mortality. What about transmission factors? Did different provinces have different methods for controlling the epidemic: such as business closures, school closures, and mask mandates, or were all these COVID social controls set exclusively at the national level with no management distinctions by province? Please comment on this in the manuscript. What about obesity and diabetes metrics? We have found that these two comorbidities greatly increase the risk for COVID admission to the ICU and mortality. Please obtain data on diabetes and obesity at the provincial level and include these important factors. As it stands, provinces that raise more pigs might be more rural, less educated, and also have higher obesity rates leading to COVID mortality rather than the NH3. These additional analyses with aggregate provincial level data would greatly strengthen the substance of this analysis and findings.

Author Response

(The authors gave the same response as above.)

Reviewer 3 Report

The current study links SARS-CoV-2 infection and death to atmospheric particulate matter and nitrogen dioxide exposure produced in agricultural activities, from March to April 2020.  The authors developed what they have called an “ecological study” that compares all-cause mortality during those two months in 2020, to the same period in the previous five years. The study allowed concluding that all pollutants together determined a 14.9% increase in mortality mainly due to NH3 emissions increment. Based on the evidence, the authors recommend NH3 exposure limits mainly when COVID-19 pandemic is active. The study addresses a topic of utmost importance, and it is very related to the mission and aims of the Journal MDPI- IJERPH.

The abstract must be improved by adding the objectives of the study. Despite the study correlates SARS-CoV-2 infection and death to atmospheric particulate matter and nitrogen dioxide exposure produced in agricultural activities in Italy during the pandemic crisis in that country from March to April 2020, no substantial pieces of evidence are shown to determine this assumption. The author must work much more on trying to set a link between NH3 emissions increment and SARS-CoV2 infection and death. It seems to exist a correlation but not a cause that determines death increment.   

On the other hand, there is no evidence in the abstract of the main goals of the study. Is it related to a general study related to infection and death determined by atmospheric particulate matter and nitrogen dioxide exposure produced in agricultural activities?  Is it part of a deeper study that analyses SARS-CoV-2 infection and death? This research must be better framed so that the main objectives of this study can sound clear.

The Introduction Section (Section 1) is very little detailed. It is missing the structure of the article, mainly the sequence that the author chose to develop the study, the name of each Section and a brief description of its content, and the novelty of this research. The authors assumed that long-term exposure to farming-related air pollutants might predispose to increased risk of COVID-19-related death, based on the fact that that COVID-19-related death is underestimated, so all-cause mortality is the epidemiologic indicator of choice for assessing the mortality impact of COVID-19. This is a hazard assumption. It must be much better grounded to allow implementing the correlation proposed by the authors in this study. Despite the weakness of these assumptions, the authors must answer the following questions: What are the main objectives of this study? What is the final goal for this research? The topic of investigation is indeed very important regarding public health control and prevention nevertheless it is not evident how do authors use these results for. How can this investigation be correlated with SARS-CoV-2  combating? The authors much work much more on that.

The authors didn't include in the research a specific section for “Literature review”.  The related works are presented in Section 1 (Introduction). The number of references (42) is short and is developed without relating the quoted studies to each other. Some new references should be added in a specific Section called “Literature review”, “Related works” or “State of the Art”. All new references should be related to the analyzed subject and it would be appreciated to quote some articles published in the International Journal of Environmental Research and Public Health regarding the investigation subject. There´s a lack of literature review for other countries' experiences concerning this analysis. It would be interesting to have this specific analysis for other places different from Italy, despite the novelty of the theme. In the last year, many scientific articles were produced related to the subject.

Section 2 regarding Materials and Methods includes subsections related to the “Study Design”, “Pollutant Exposure”, “Meteorological Variables”, “outcome” and “Statistical Methods”. The ecological study is very well detailed, however, is not clear what is the influence of farming-related air pollutants exposure to SARS-CoV-2 infection and death. What similar studies worldwide were already developed? What other variables must be controlled, besides meteorological variables?

Section 3 and 4 refer to results presentation and discussion. This study presents many strengths since it is very analytical research, however, there is some missing information that makes this investigation incomplete. The data presented in Tables 1 and 2 must be better explained since no extra analysis is done, and no correlation of this investigation with other variables besides meteorological variables is set. Some extra discussion on the results must be worked out. The authors are invited to work much more on results discussion, adding some new blocks of text and some extra graphs and charts are preferable to organize information and better explain obtained correlations. It is of utmost importance to explain why and how there is a link between SARS-CoV-2 infection, environmental degradation, and climate change.

Section 4 is the conclusion section. Indeed, it is very poor. Conclusions must synthesize thoroughly all the outcomes of the investigation. Conclusions are not mere findings. I repeat, some stronger and meaningful conclusions are needed. The authors say that intensive agriculture is a major cause of biodiversity loss worldwide, and has been implicated in the emergence and dissemination of SARS-CoV-2 and other viruses. It is evident that this is true, however, this study doesn´t show how this happens. This is a robust ecological study that was not properly worked and correlated. The authors must do it more profoundly.

Author Response

(The authors gave the same response as above.)

Round 2

Reviewer 1 Report

The author basically answered all my doubts. Personally, it is recommended to accept it for further publication.